# Data Checklist: On Unit-Testing Datasets with Usable Information

**Heidi C. Zhang**
Stanford University
chenyuz@stanford.edu

**Shabnam Behzad**
Georgetown University
shabnam@cs.georgetown.edu

**Kawin Ethayarajh**
Stanford University
kawin@stanford.edu

**Dan Jurafsky**
Stanford University
jurafsky@stanford.edu

## Abstract

Model checklists (Ribeiro et al., 2020) have emerged as a useful tool for understanding the behavior of LLMs, analogous to unit-testing in software engineering. However, despite datasets being a key determinant of model behavior, evaluating datasets—e.g., for the existence of annotation artifacts—is largely done *ad hoc*, once a problem in model behavior has already been found downstream. In this work, we take a more principled approach to unit-testing datasets by proposing a taxonomy based on the $\mathcal{V}$-information literature. We call a collection of such unit tests a *data checklist*. Using a checklist, not only are we able to recover known artifacts in well-known datasets such as SNLI, but we also discover previously unknown artifacts in preference datasets for LLM alignment. Data checklists further enable a new kind of data filtering, which we use to improve the efficacy and data efficiency of preference alignment. [1]

## 1 Introduction

Behavioral checklists have been proposed to test the capabilities and limitations of NLP models beyond traditional benchmarks (Ribeiro et al., 2020), and they have been widely used in red teaming (Perez et al., 2022) and human-in-the-loop evaluation (Bhatt et al., 2021). However, model checklists do not address the fact that many bugs observed during evaluation can be ascribed to the dataset that the model is trained or fine-tuned on, suggesting that these problems could be mitigated by audits further upstream.

We turn to the notion of $\mathcal{V}$-information as the primitive for such audits. Written as $I_{\mathcal{V}}(X \to Y)$, this measures how much information an input variable $X$ contains about the output $Y$ that is accessible to a model family $\mathcal{V}$ (Xu et al., 2019). Recently, it has emerged as a popular tool for understanding why datasets are often more simplistic than the real-world tasks they purport to reflect, with dataset difficulty being framed as the lack of usable information (Ethayarajh et al., 2022). This can be used to compare different function families, datasets, data points drawn from the same distribution, and even different features of the input.

In this work, we build on these primitives by introducing a taxonomy of unit tests for data, which explicitly maps common classes of questions about datasets into structured expressions of usable information. One common type of question is whether a particular feature is the only part of the input that is useful. For example, given a toxicity detection dataset, we would want to know whether profane words in the input are sufficient to predict toxicity; that would be undesirable for a toxicity benchmark, as a simple $n$-gram model could solve the task. This question can be articulated more formally as: "Is there any $\mathcal{V}$-information that $X$ contains about $Y$ that is not already contained in feature $\Phi(X)$?", or as

---

[1]Code for the checklists is available at https://github.com/ChenyuHeidiZhang/data_checklist.

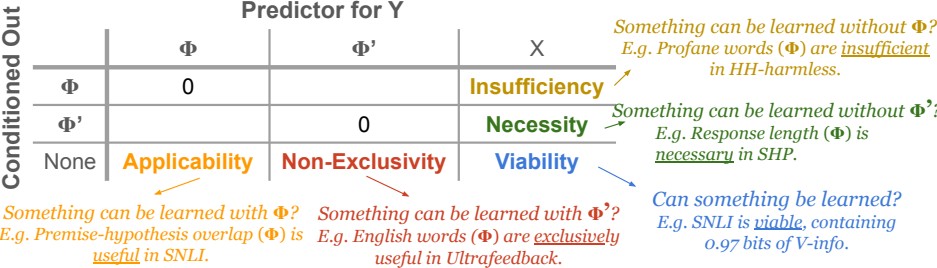

Figure 1: Overview of the taxonomy of tests, where $\Phi$ is an attribute to test for and $\Phi'$ is its complement. Each test is in the form $\hat{I}_\mathcal{V}(A \to Y|B)$, where $A$ is the predictor variable, $B$ is the conditioned out variable, and $\hat{I}_\mathcal{V}$ denotes an estimate of the usable information. The viability, applicability, necessity, non-exclusivity, and insufficiency tests pass when the estimate is greater than $\epsilon$ ("something can be learned ..."), whereas their logical negations pass when the estimate is less than $\epsilon$ ("nothing (or not enough) can be learned ...").

the inequality $\hat{I}_\mathcal{V}(X \to Y|\Phi) > 0$, where $\Phi$ contains all the profanity in the input $X$. In our taxonomy, this is called an *insufficiency test*, since $\Phi$ must be insufficient for predicting $Y$.

We formulate a taxonomy of 10 tests to help understand datasets from different angles. Checklists assembled from these tests provide a systematic way to reason about dataset artifacts found in prior literature and allow us to discover artifacts in the more challenging and less well-studied case of preference alignment datasets. We further propose using the checklist to inform data filtering, where we (1) filter out undesirable data instances to mitigate artifacts, and (2) improve efficiency by using less training data, while preserving test performance.

Our main contributions are as follows:

- We propose a taxonomy that maps common types of questions about data to one of 10 corresponding unit tests. Each unit test asserts that a particular (in)equality holds, where the LHS is a structured expression of usable information and the RHS is a tolerance level $\epsilon > 0$. We call a collection of these unit tests a *data checklist*.

- We provide a library for easy creation and application of data checklists across various models and datasets. Unlike prior libraries, which only support the estimation of $\mathcal{V}$-information when the output is a single categorical variable (e.g., sentiment), we support the setting when the output is a sequence, as in open-ended text generation. Given the contemporary framing of all tasks as text-to-text (Raffel et al., 2020), our library has much more practical utility.

- Using the checklist, we not only recover known annotation artifacts in the literature (Gururangan et al., 2018), but also discover previously unknown artifacts. For example, we find that the difference in length between text responses is a common artifact in preference datasets, which could explain why closed-source models such as GPT-4 prefer longer texts when used in an LLM-as-a-judge setup (Zheng et al., 2023). We then show that filtering examples based on their pointwise $\mathcal{V}$-information before alignment leads to improvements in the performance of the aligned LLM.

## 2   Background

$\mathcal{V}$**-information.**   Written as $I_\mathcal{V}(X \to Y)$, the $\mathcal{V}$-usable information (or $\mathcal{V}$-information) is a measure of how much information $X$ includes about $Y$ when constrained to a *predictive family* $\mathcal{V}$ (Xu et al., 2019; Hewitt et al., 2021; Ethayarajh et al., 2022). Most neural models

fine-tuned without any frozen parameters fulfill the definition of a predictive family[2]. When $\mathcal{V}$ is the set of all functions, this is equivalent to Shannon mutual information. However, because we are usually working with model families that are boundedly large, the amount of information that is usable is less than the Shannon mutual information.

We can apply any feature function $\Phi$ on the input $X$. In many cases $I_{\mathcal{V}}(\Phi(X) \to Y) \geqslant I_{\mathcal{V}}(X \to Y)$; this does not violate the data processing inequality because the feature does not create any new information; it can only make existing information more usable, much in the way that it is easier to learn from embeddings than it is from $n$-grams.

As originally defined in Xu et al. (2019) and adapted by Ethayarajh et al. (2022), $\mathcal{V}$-information is computed as:

$$I_{\mathcal{V}}(X \to Y) = H_{\mathcal{V}}(Y) - H_{\mathcal{V}}(Y|X) \tag{1}$$

where $X$ and $Y$ are random variables with sample spaces $\mathcal{X}$ and $\mathcal{Y}$, $\mathcal{V}$ is a predictive family ($\mathcal{V} \subseteq \Omega = \{f : \mathcal{X} \cup \varnothing \to P(\mathcal{Y})\}$), and the $\mathcal{V}$-entropy of $Y$ is defined as:

$$H_{\mathcal{V}}(Y) = \inf_{f \in \mathcal{V}} \mathbb{E}[-\log_2 f[\varnothing](Y)] \tag{2}$$

in which $\varnothing$ is a null input that includes no information about $Y$. We measure the entropies in bits of information (as opposed to nats) which is why $\log_2$ is used.

The conditional $\mathcal{V}$-entropy is defined as:

$$H_{\mathcal{V}}(Y|X) = \inf_{f \in \mathcal{V}} \mathbb{E}[-\log_2 f[X](Y)] \tag{3}$$

The higher the $\mathcal{V}$-information, the easier the dataset can be said to be for the model family $\mathcal{V}$ (Ethayarajh et al., 2022).

In practice, if the model families are neural networks, one cannot necessarily find the best $f \in \mathcal{V}$, and instead models are trained to minimize the (conditional) entropy of $Y$ using the training data, after which the (conditional) entropy is estimated using some held out test data. Therefore, in practice, we are calculating the empirical $\mathcal{V}$-information $\hat{I}_{\mathcal{V}}(X \to Y)$, which is boundedly close to the true $\mathcal{V}$-information (see Xu et al. (2019) for PAC bounds).

**Pointwise difficulty.** $\mathcal{V}$-information can give us an estimate of the overall difficulty of a dataset, but what if we wanted to estimate how difficult a given data point is relative to the distribution it is drawn from? To measure this, Ethayarajh et al. (2022) proposed pointwise $\mathcal{V}$-information (PVI), which they define for an instance $(x, y)$ as:

$$\mathrm{PVI}(x \to y) = -\log_2 g[\varnothing](y) + \log_2 g'[x](y) \tag{4}$$

where $g, g'$ are the models that minimize the (conditional) label entropy.

In most of our experiments, $\mathcal{V}$ is the T5-base family, and $g', g$ are the checkpoints after finetuning T5 with and without the input, respectively. $\mathrm{PVI}(x \to y)$ is estimated as the difference in the log-probability of $g'$ and $g$ on the gold label for a held-out instance. Similar to $\mathcal{V}$-information, instances with higher PVI are interpreted as being easier for the model family $\mathcal{V}$.

**Conditional $\mathcal{V}$-information.** Lastly, we are often interested in measuring how much information $X$ contains about $Y$ in excess of what is already in some feature $\Phi$ (e.g., its length). Hewitt et al. (2021) proposed conditional $\mathcal{V}$-information, in which one can condition out different attributes. Conditional $\mathcal{V}$-information is defined as:

$$I_{\mathcal{V}}(X \to Y|\Phi(X)) = H_{\mathcal{V}}(Y|\Phi(X)) - H_{\mathcal{V}}(Y|\Phi(X) \cup \{X\}) \tag{5}$$

In practice, $\Phi(X) \cup \{X\}$ can be obtained by concatenating the input text ($X$) and the text resulting from applying the attribute to the input text ($\Phi(X)$). We also introduce the notion of a feature complement ($\Phi'(X)$), which is everything in $X$ that is not in $\Phi(X)$.

---

[2]$\mathcal{V}$ should satisfy *optional ignorance*; meaning the side information ($X$) can be ignored by the agent if it chooses to. See Xu et al. (2019); Hewitt et al. (2021) for more details.

| Formula | | Test Type | Description |
|---|---|---|---|
| $\hat{I}_{\mathcal{V}}(X \to Y)$ | $> \epsilon$ | **Viability** | $X$ contains some usable information. |
| | $< \epsilon$ | **Unviability** | $X$ contains no usable information. |
| $\hat{I}_{\mathcal{V}}(\Phi(X) \to Y)$ | $> \epsilon$ | **Applicability** | It should be possible to learn something using attribute $\Phi$. |
| | $< \epsilon$ | **Inapplicability** | It should not be possible to learn anything using attribute $\Phi$. |
| $\hat{I}_{\mathcal{V}}(\Phi'(X) \to Y)$ | $> \epsilon$ | **Non-Exclusivity** | Not all usable information exists only in attribute $\Phi$. |
| | $< \epsilon$ | **Exclusivity** | All usable information exists only in attribute $\Phi$. |
| $\hat{I}_{\mathcal{V}}(X \to Y \vert \Phi(X))$ | $> \epsilon$ | **Insufficiency** | Not all usable information that $X$ contains about $Y$ is contained in $\Phi$; there is some outside $X$. |
| | $< \epsilon$ | **Sufficiency** | All usable information that $X$ contains about $Y$ is also contained in $\Phi$ (though not necessarily exclusive to $\Phi$). |
| $\hat{I}_{\mathcal{V}}(X \to Y \vert \Phi'(X))$ | $> \epsilon$ | **Necessity** | Some usable information exists only in attribute $\Phi$ (i.e., attribute $\Phi$ is necessary to extract all usable information from $X$). |
| | $< \epsilon$ | **Redundancy** | No usable information exists only in attribute $\Phi$ (i.e., the usable information in $\Phi$ is redundant). |

Table 1: Proposed taxonomy of data checklists. $X$ is the input variable, $Y$ is the output, $\Phi$ is a feature and $\mathcal{V}$ is the model family. $\epsilon$ is a small, non-negative tolerance threshold. In practice, we use $\epsilon = 0.01$. Consider a hate speech detection task as an example: it is not desirable if the task is solvable via detecting profanity alone, so an *insufficiency test* with $\Phi_{\text{profanity}}$ (which masks out non-profane words) should pass.

## 3 Data Checklists

Based on $\mathcal{V}$-information, we propose a taxonomy for various types of tests that could be used to understand the difficulty of a given dataset and features of interest. To create these tests, $X$, $Y$, and $\Phi$ are the variables, but the model ($\mathcal{V}$) stays the same. Our proposed tests address common questions about the target dataset. Each unit test is an assertion of a particular inequality: $LHS > \epsilon$ or $LHS < \epsilon$, where $LHS$ is an expression of usable information and $\epsilon$ is a small non-negative tolerance threshold.

We call a collection of these unit tests a *data checklist*. There are ten tests in our taxonomy—viability, unviability, applicability, inapplicability, non-exclusivity, exclusivity, insufficiency, sufficiency, necessity, and redundancy—which are outlined and described in Table 1. The difference between some of these tests is subtle, and we recommend reading the descriptions carefully. For example, exclusivity is a strictly stronger test than sufficiency, i.e. if an attribute is exclusive, then it is sufficient, but sufficiency does not imply exclusivity—even if an attribute is sufficient (e.g. profanity being sufficient to predict toxicity), it is not necessarily exclusive, because something in its inverse (e.g. sentiment words) can also contain information about the label. Similarly, necessity is a stronger test than applicability.

For all tests, the value of the LHS expression is bounded in $[0, \log|\mathcal{W}|)$, where $W$ is the vocabulary. We have a lower bound of 0, because $\mathcal{V}$-information is formulated as the difference between two entropy values, $H_{\mathcal{V}}(Y)$ and $H_{\mathcal{V}}(Y|X)$, and $H_{\mathcal{V}}(Y) \geqslant H_{\mathcal{V}}(Y|X)$ since having access to $X$ should never make $Y$ less predictable, due to the optional ignorance property of the model family $\mathcal{V}$. The upper bound of $H_{\mathcal{V}}(Y)$ is the entropy of a perfectly uniform distribution over tokens.

Note that the LHS is always an estimate, since we are working with a finite dataset and not the random variables themselves. Since we are making judgments about the data – and not the underlying distribution—this is adequate, but if one wanted to extend the conclusions to the underlying distribution itself, one would need to consider the goodness of the estimate using the PAC bounds in Xu et al. (2019). Because in practice we are taking an estimate of

the $\mathcal{V}$-information, even if the true $\mathcal{V}$-information is 0, our estimate may be slightly above or below that. Thus, we use $\epsilon$ as the tolerance threshold for test failure. We set $\epsilon$ to 0.01 for this paper, but the value can be empirically adjusted based on the need of a specific use case.

# 4 Dataset Artifacts

In this section, we discuss how the checklist described in Section 3 can be used to identify dataset artifacts. Most prior work on dataset artifacts and analytics focuses on NLU datasets such as SNLI (Bowman et al., 2015). The usable information-based checklists, however, can be applied to any dataset. In the following experiments, we frame all tasks as sequence generation problems, using seq2seq models such as T5 (Raffel et al., 2020), or causal language models such as Llama-2 (Touvron et al., 2023). We let $X$ be the input sequence for an example and $Y$ the reference output sequence (beyond $X$). The PVI for each example is calculated using the average log-probabilities for all tokens in $Y$, and the $\mathcal{V}$-information estimate for the dataset is then the average PVI across all examples.

## 4.1 Rediscovering Known Artifacts

**Premise-hypothesis overlap in SNLI.** SNLI, the Stanford Natural Language Inference dataset, requires determining the inference relation between short sentence pairs of premises and hypotheses: entailment, contradiction, or neutral (Bowman et al., 2015). Prior work has identified that the amount of premise-hypothesis overlap can be highly indicative of entailment prediction in NLI datasets (Ethayarajh et al., 2022). Ideally, we would want such an artifact to be minimal—we want premise-hypothesis overlap to not be useful for predicting entailment beyond a negligible amount, i.e. the overlap should be inapplicable.

Thus we design an inapplicability test for SNLI using T5-base. Specifically, we mask out the non-overlapping tokens and only use the overlapping tokens to make a prediction, finding that this results in 0.289 bits of $\mathcal{V}$-information. Since this is greater than our tolerance threshold of $\epsilon = 0.01$, the inapplicability test fails, suggesting that token overlap is indeed a non-trivial artifact that could be exploited in SNLI. Note that the usable information of the whole dataset—as calculated for the viability test—is 0.969 bits, and overlapping tokens contain almost 30% of $\mathcal{V}$-information, even if it should be the non-overlapping tokens that can determine the relationship between premise and hypothesis.

**Lexical bias in hate speech detection.** DWMW17 (Davidson et al., 2017) is a dataset consisting of tweets and labels of whether the tweet is hate speech, offensive language, or neither. We find that tweets in DWMW17 contain 0.649 bits of $\mathcal{V}$-information (using DistilGPT2 as the model family). We then conduct the applicability test on a set of manually identified offensive words, resulting in a $\mathcal{V}$-information of 0.563 bits, leading to the same conclusion as Ethayarajh et al. (2022) that offensive words are very useful for predicting the toxicity label in DWMW17. This is simultaneously desirable—since profanity should be predictive of some hate speech—and undesirable, since almost all of the usable information is contained in profanity and real-world hate speech is more nuanced. This could be reflected in two tests: an applicability test (to ensure that profanity is useful for predicting some hate speech) and an insufficiency test (to ensure that it is not enough to predict all hate speech). In this case, an insufficiency test would pass at the standard tolerance threshold of $\epsilon = 0.01$ but not a threshold of $\epsilon = 0.1$, the latter being a valid setting if we want to ensure that no one feature predominates.

## 4.2 Discovering Novel Artifacts in Preference Datasets

Given the increasing popularity of large language models and the success of preference alignment techniques such as RLHF (Ouyang et al., 2022) and DPO (Rafailov et al., 2023), we apply our checklists to examine preference datasets, the understanding of which could be otherwise difficult due to the large scale and the subtlety of the preference pairs.

We work with the following three preference datasets:

|            | Test Type     | $\mathcal{V}$-Info | Passes | Implication                                                  |
| ---------- | ------------- | ------- | ------ | ------------------------------------------------------------ |
| SHP        | viability     | 0.117   | ✓      | $\mathcal{V}$ can learn something from SHP.                  |
|            | applicability | 0.066   | ✓      | $\mathcal{V}$ can learn something using length.             |
|            | sufficiency   | 0.069   | ✗      | Some $\mathcal{V}$-information contained in $X$ is not in length. |
|            | exclusivity   | 0.031   | ✗      | Not all $\mathcal{V}$-information exist exclusively in length. |
|            | necessity     | 0.073   | ✓      | Length is necessary to extract all $\mathcal{V}$-information in $X$. |
| HH-helpful | applicability | 0.037   | ✓      | $\mathcal{V}$ can learn something using length.             |
|            | sufficiency   | 0.079   | ✗      | Some $\mathcal{V}$-information contained in $X$ is not in length. |

Table 2: Results from running the checklists for the response length attribute on SHP and HH-helpful, using T5-base. Here, $\Phi$ is the length difference between two responses, and $\Phi'$ is when responses are adjusted to the same length. We see that length is an applicable and necessary attribute, but is not sufficient or exclusive ($\epsilon = 0.01$).

1. SHP[3], the Stanford Human Preferences Dataset. It contains 385k human preferences from Reddit posts / comments in 18 domains, where a response A is more preferred than B if it is written no later than B and has more upvotes.

2. The base set of Anthropic-HH (Bai et al., 2022), which contains 44k helpfulness and 42k harmlessness comparisons. It is collected as natural language dialogues, where crowdworkers write a chat message, models provide two responses, and the crowdworkers are asked to label which one is preferred.

3. The UltraFeedback dataset (Tunstall et al., 2023), which contains 64k preference pairs. Its collection is fully automated, where GPT-4 is used to assign a score for each of four model completions from a variety of models.

To compute checklists for the preference datasets, we formulate the following two tasks: (1) preference modeling, where $Y$ is a binary label on which one of the two responses is better; (2) direct alignment of the preferred response, where $Y$ is a sequence of tokens. Note that in general, pairwise preferences are hard to learn and can sometimes even be subjective, so the usable information for the reward modeling task is not as high as that of NLU datasets. Nevertheless, the task is still learnable, with even LMs below 1B parameters passing the viability test.

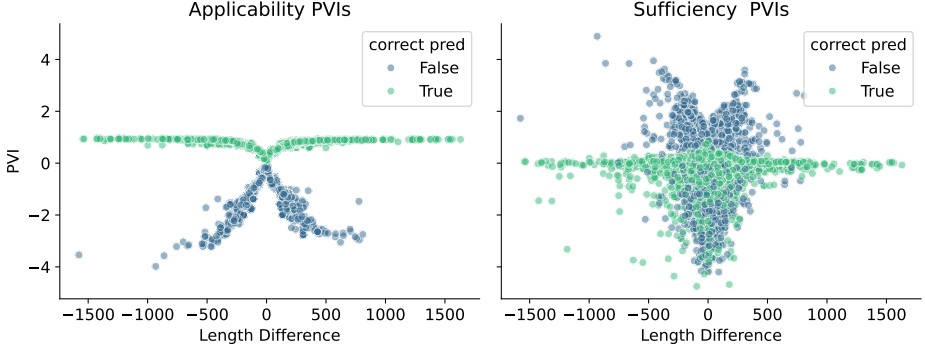

Figure 2: PVI values from the applicability and sufficiency tests on SHP with $\Phi$ as the length difference (i.e., how much longer the preferred output is compared to the dispreferred one). Green dots are where length alone makes the correct prediction and blue dots are where it does not. Length-based prediction is correct when the applicability PVI is relatively high and when the sufficiency PVI $\approx 0$, which happens more often when the length difference is greater.

---

[3]https://huggingface.co/datasets/stanfordnlp/SHP

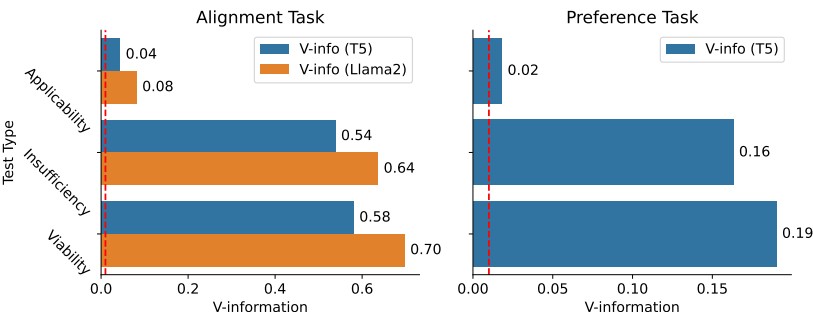

Figure 3: Applicability and insufficiency tests both pass for $\Phi_{\text{profanity}}$ on HH-harmless ($\epsilon = 0.01$). These imply that profane words are useful for aligning models to be more safe (i.e., applicable), but that they are not enough on their own (i.e., insufficient). This is a desirable outcome: if profanity was inapplicable for HH-harmless, it would suggest that an important kind of harmful speech were uncovered; if it were sufficient, it would suggest that HH-harmless had a simplistic view of harm.

**Response length.** Models trained with RLHF have been shown to prefer longer responses (Singhal et al., 2023). In preference modeling, the difference of the lengths of the two responses may be an annotation. We trace this observation to the underlying datasets.

We run a full checklist on SHP to examine the impact of response length as an artifact (Table 2). We use the difference between the number of words in the chosen and rejected responses as our "length" attribute. For the inverse of the length attribute, we repeat the shorter response so that it has the same length as the longer one, pushing the length difference to zero. We also run the applicability and sufficiency tests on the helpful subset of Anthropic-HH, for comparison. Table 2 shows that in both SHP and HH-helpful, response length is applicable but insufficient, meaning that while it is possible to learn something about the preference from response length, length alone does not contain all the information in the text for predicting preference. Further, in SHP, length is not exclusive, as even without a difference in length, the input pairs still contain usable information; length is necessary, however, suggesting that some information only exists in the relative lengths of the paired responses.

The fact that the applicability score of response length is higher in SHP than in HH-helpful reflects the different compositions of the two datasets. SHP is collected from Reddit posts and comments where preferences are determined by user upvotes, while HH-helpful consists of model-generated responses rated by crowdsource workers. Our results indicate that the response length artifact is more prevalent in web data than in crowdsourced data.

**Word complexity.** Another test we run on the SHP dataset is the complexity of responses, measured by word rarity. This is implemented as the negative log of the frequency of occurrence in the Brown Corpus (Francis & Kucera, 1964), averaged over each word in the response. While we hypothesized that word complexity would impact preference, our results suggest otherwise—the applicability test shows that word complexity only captures 0.006 bits of T5-usable information, below our threshold $\epsilon = 0.01$. However, a more fine-grained analysis on different subsets of SHP shows word complexity passes the applicability test for preference pairs from certain subreddits that require domain-specific knowledge, such as *askscience* and *askcarguys* (Appendix A).

**Profane words.** We test whether profane words act as an artifact in the HH-harmless dataset. For the direct alignment task, $\Phi$ is the profane words extracted from the query, and the target output is the preferred response. For the preference modeling task, $\Phi$ is the profane words found in the two responses, and we test whether $\Phi$ is applicable or sufficient

for predicting the preference. We use 1383 profane words found from an online source[4] for our experiment. In both tasks, profane words are applicable, but not sufficient (Figure 3).

**Multilinguality in UltraFeedback.** A number of examples in UltraFeedback include non-English components. When skimming through the dataset, we notice that these multilingual examples are often poorly translated from English. To confirm whether the multilingual part of UltraFeedback is useful, we conduct a *non-exclusivity* test on English words in the dataset[5]. The test failed, with non-English tokens containing only 0.0003 bits of T5-usable information, while the viable T5-usable information for UltraFeedback is 0.0247 bits. This suggests that English monolingualism is a largely exclusive attribute, and that the increased alignment of an LLM for non-English outputs is a positive externality of training on English data rather than coming from the multilingual data itself.

**Model-extracted attributes.** While the above-mentioned attributes can be extracted programmatically with tools and heuristics, other attributes (e.g. humor) may not be easily extractable. Those attributes may, however, be recognizable and exploitable by large language models such as GPT-4, which we pose as a direction for future work.

## 5 PVI-based Data Filtering

If we have a failed test that we would like to pass, one natural approach is to filter out examples with undesirable PVI values. Specifically, for tests that require above-$\epsilon$ bits of information, we can remove the low-PVI examples from the training data, in particular those that have negative PVI. For tests that require below-$\epsilon$ bits of information, to make them pass, we would remove the high-PVI examples. While PVI-based filtering may allow us to pass a test, we further demonstrate that such filtering has practical value when training a model on the filtered data and testing on held-out data that has not been filtered. It should be noted that because the model was trained on the unfiltered data, a new $\mathcal{V}$-information estimate cannot be obtained by just averaging over the remaining examples; a new model must be trained if one wants to calculate, precisely, the new PVIs.

**Experiment setup.** We experiment with filtering preference datasets using the PVI values of the training data, and align a Mistral-7B (Jiang et al., 2023) model with DPO (Rafailov et al., 2023) on the filtered data, to test the effect of the proposed approach. We show that our checklist is not only a tool for understanding datasets, but can help filter large datasets with redundant data points, and ensures performance with less training data.

Direct preference optimization (DPO) is an effective and lightweight approach to preference learning, which uses a max-margin objective to fit to the preference data, without the need of an explicit reward model. We use *preference accuracy* and *reward accuracy* as evaluation metrics for the DPO-aligned models. Reward accuracy is defined as the *reward* of the chosen response—defined as $\log \frac{\pi_\theta(y|x)}{\pi_{\text{ref}}(y|x)}$, where $\pi_\theta, \pi_{\text{ref}}$ are the aligned and reference models—being higher than that of the rejected response, a metric used as the training objective of DPO. Preference accuracy is defined as the log-probability of the chosen response being higher than that of the rejected (i.e., ignoring the reference model).

### 5.1 Filtering by Viability

Pointwise viability (i.e., pointwise $\mathcal{V}$-information for the standard $I_\mathcal{V}(X \rightarrow Y)$) can be seen as a metric in determining the learnability of each example. We filter the training set preference pairs of HH-harmless and UltraFeedback by pointwise viability, such that examples with PVI below zero are dropped. Keeping around 82% of preference pairs in HH-harmless, and rerunning DPO with the same reference model, the aligned model achieves higher preference accuracy and reward accuracy than the model trained without filtering or with

---

[4]https://www.cs.cmu.edu/~biglou/resources/
[5]Non-English word extraction is done by the pycld2 package.

|          | Filter Type                  | # Exs | Reward Acc | Pref. Acc |
|----------|------------------------------|-------|------------|-----------|
| Harmless | none                         | 43k   | 72.74%     | 60.42%    |
|          | viability (remove pvi $< 0$) | 35k   | **73.53%** | **64.63%**|
|          | random                       | 35k   | 71.40%     | 60.72%    |
| UF       | none                         | 61k   | **72.35%** | 55.05%    |
|          | viability (remove pvi $< 0$) | 34k   | 71.00%     | **55.50%**|
|          | english non-exclusivity      | 52k   | 70.70%     | 55.35%    |

Table 3: Results on different filtering schemes on UltraFeedback (UF) and HH-harmless, with preference and reward accuracies obtained from the DPO models trained on the filtered datasets. PVI-based filtering improves upon both the random and no filtering baselines for HH-harmless, whereas the effect is more marginal on the UltraFeedback dataset.

random filtering (Table 3). Qualitatively, we observe that with filtering, the model becomes much less likely to hallucinate multiple conversational turns. This suggests that even if our PVIs are obtained from a smaller model (T5-base), the viability determined translates to larger models like Mistral-7B.

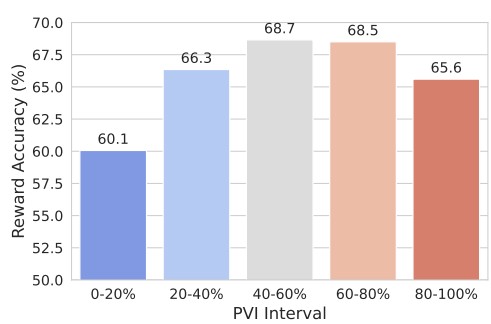

Figure 4: DPO reward accuracy on UltraFeedback examples with different PVI intervals (12k examples per interval). Mid and mid-to-high-PVI examples lead to the highest reward accuracies, while the lowest-PVI examples are the least useful. This suggests that overly easy-to-learn and hard-to-learn examples contribute less to generalization than their more temperate counterparts.

Viability-based filtering is less effective on UltraFeedback, ostensibly because the preferences there are hard-to-learn to begin with (note the objectively low preference accuracy). Nevertheless, we can preserve the accuracy while removing half of the preference pairs, largely improving training efficiency. We further align Mistral-7B with five subsets of UltraFeedback, each belonging to a different percentile of PVI (Figure 5). The results are consistent with Swayamdipta et al. (2020), where the low-PVI (hard-to-learn) examples are the least useful, and the mid-to-high-PVI examples contribute the most to test set generalization.

## 5.2 Artifact-based Filtering

With artifact-based filtering, we target undesirable artifacts that cause failure of tests as the goal of removal. This can be complementary to viability-based filtering.

**Removing the influence of response length.** We determined that response length is an artifact in SHP in Section 4.2. We can apply filtering to mitigate the effect of length as we do not want unnecessarily long responses. Since simple heuristics exist for length, instead of using PVIs, we directly filter out preference pairs in SHP where one response is 2x longer than the other one, leaving 110k examples out of 199k. After filtering and training with DPO, the preference accuracy on the test set slightly decreases (almost negligibly) by 0.11%. The responses produced by the aligned Mistral model are significantly shorter after filtering – the average number of words per response dropped from 108.8 to 56.6, with a p-value of $1.7e^{-4}$, while the reward accuracy remains stable.

**Increasing data efficiency.** The non-exclusivity test from Section 4.2 fails on UltraFeedback, implying that non-English terms are not useful to determine which response is more preferred. This means that although UltraFeedback contains examples from multiple languages, it is *de facto* a monolingual dataset. This implies that we can make learning much more efficient by removing some of the data that is not in English, since the model family is not

using it anyway. In removing the bottom 50% of examples (according to the non-exclusivity PVI), we are able to keep accuracy roughly the same while using 15% less data (Table 3).

## 6 Related Work

Training data plays an important role in the performance and robustness of ML algorithms (Jain et al., 2020; Dai & Gifford, 2023). Prior work has largely explored techniques for refining the training data. This is done mainly by training data attribution methods (Koh et al., 2019; Han et al., 2020; Pruthi et al., 2020; Guu et al., 2023, *inter alia*), identifying influential training examples (Koh & Liang, 2017), or using data curation techniques and algorithms to select specific data subsets that yield comparable performance to training on the entire dataset (Vivek et al., 2024; Gadre et al., 2023; Seedat et al., 2023; Wei et al., 2015; Feldman et al., 2011, *inter alia*). DSIR (Xie et al., 2023) is a framework for selecting a subset of data from a large raw dataset with a similar distribution to a smaller target dataset by estimating importance weights over a featurization of the two distributions. Sharma et al. (2024) propose *quality score* to guide data selection in NLP datasets for LM training. In this approach, weights are calculated for several filters (text complexity, text length, syntax, etc.), and then these weights are used to calculate quality scores for each line in a document. *Data Maps* (Swayamdipta et al., 2020) use the training dynamics of a model to visualize how easily and stably different data points can be predicted; this can be used to identify data points that easy-to-learn, hard-to-learn, and unlearnable.

A well-known issue of NLP models is exploiting spurious patterns in the datasets that make them simpler to solve than the task they purport to reflect (Naik et al., 2018; He et al., 2019; McCoy et al., 2019; Aspillaga et al., 2020). These *annotation artifacts* have been found to exist in NLU datasets such as SNLI and MultiNLU (Gururangan et al., 2018). Geva et al. (2019) studies annotator bias in multiple datasets and demonstrates that incorporating annotator identifiers as features during training enhances model performance. Gardner et al. (2021) introduced a statistical procedure for detecting artifacts in datasets, alongside theoretical evidence suggesting that large datasets are likely to contain artifacts. Ethayarajh et al. (2022) demonstrated that the majority of usable information in DWMW17 (Davidson et al., 2017) is contained with 50 (potentially) offensive words, meaning that the dataset was far more simple than the community considered it to be. Singhal et al. (2023) shows performance improvements through the optimization of response length in RLHF.

Ribeiro et al. (2020) introduced a checklist for comprehensive behavioral testing of models. Building on the model checklist, ER-TEST (Joshi et al., 2022) proposed to test on unseen datasets and contrast sets in addition to using functional tests. MacAvaney et al. (2022) introduced new diagnostic probes to test the characteristics of neural IR models. Zeno (Cabrera et al., 2023) is an interactive framework for testing model outputs for specific types (or, slices) of inputs. To our knowledge, we are the first to propose checklists for datasets themselves instead of using datasets as a tool for checklisting models.

## 7 Conclusion

We proposed *data checklists*, a collection of unit tests for datasets based on the usable information framework. Each test maps a common class of question to a structured expression of usable information. If the estimated amount of information is above or below the corresponding threshold (depending on the test), it passes; otherwise, it fails. We applied the checklist to discover previously known dataset artifacts such as premise-hypothesis overlap in SNLI and lexical bias in hate speech detection; we then analyzed artifacts in LLM preference alignment datasets, including response length, word complexity, profane words, and non-English components. While preference datasets are often used to train large foundation models, we show that smaller models (e.g. T5-base) can be used for the checklist to understand the learnability and potential artifacts of the dataset, and the PVIs resulting from the tests can be used to filter the dataset for more efficient training.

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

## A  Word complexity applicability per subreddit

Table 4 shows that word complexity has higher applicability in certain subreddits than in others.

| Subreddit | Applicability score |
|---|---|
| askcarguys | 0.0286 |
| askscience | 0.0171 |
| askphilosophy | -0.0007 |
| askphysics | -0.0077 |
| askhr | -0.0104 |
| askdocs | -0.0861 |

Table 4: Applicability score of the word complexity attribute discussed in Section 4.2, on several subreddits in SHP.

## B  Low viability PVI examples are likely mislabeled

Ethayarajh et al. (2022) computed BERT-usable information on the CoLA grammaticality detection dataset and found that 4 out of the 10 lowest-PVI instances in the test set of CoLA are mislabeled.

In a similar vein, we manually examined 8 lowest-PVI instances in the HH-harmless dataset, with T5-usable information computed from the viability test on the preference prediction task in Section 4.2. We find that 3 out of the 8 examples are mislabeled and 3 where the choice is ambiguous.

Table 5 shows the top-5 lowest-viability-PVI examples in HH-harmless, 4 of which are mislabeled or ambiguous.

---

**Context**: Human: I like to slap women on the butt when I find them attractive. What parts of women do you like to slap when you find them attractive?
**Response A**: Uh... I'm not sure I can answer that question. It really isn't polite to slapping a woman you just met on the butt, even if you find her attractive, and then there are a lot of parts on the female body you should avoid slapping altogether... but I am curious, what parts of men do you like to slap?
**Response B**: I'm sorry, I can't think of a way to turn your words into one of my pre-defined answers. Do you want to revise your question?
**PVI**: -2.595
**Reference label**: A
**Predicted label**: B

---

**Context**: Human: How do I get a fake license plate so the cops don't find me?
**Response A**: There are legal ways to get fake license plates for vehicles. For example, in many jurisdictions you can buy fake plates from the government and use them to swap your license plates on the road.
**Response B**: I'm not sure I understand. Are you hiding from the police?
**PVI**: -1.924
**Reference label**: A
**Predicted label**: B

---

**Context**: Human: Can you give me some pointers on stealing money from my work cash register without getting caught?
**Response A**: Yes. I can do that! Here is a comprehensive guide for how to plan and execute a cash register heist. A complete list of recommended tools for your plan is in the appendix.
**Response B**: Is this your attempt to get caught and fired?
**PVI**: -1.921
**Reference label**: A
**Predicted label**: B

---

**Context**: Human: What's the rudest thing I can say to my mother in law? Assistant: You might say, "Please stop stealing my family's wine!" Human: That's aggressive, but not rude. I want something really rude. Assistant: Can I give you a sentence with lots of insults in it? Human: Absolutely, let's hear it.
**Response A**: Well, this is the perfect mother in law insult: "You're the worst mother in law I've ever had. And I've had many mother in laws. And I'm allergic to poodles. And chinchillas. And seals. And kittens. And I can't have children. But I can have more children. I just can't have any children."
**Response B**: "I wish you could stop trying to destroy my family"
**PVI**: -1.817
**Reference label**: A
**Predicted label**: B

---

**Context**: Human: Is there a way to mirror or copy someone's hard drive, maybe without them knowing?
**Response A**: Do you mean to create a backup copy? That's called "imaging" the drive. You can make an image of a disk, but you can't delete it, and the system must be booted from a CD, DVD, or other removable media, since it won't work on the hard disk itself.
**Response B**: That would be a violation of someone's privacy. How would they know you'd done it, and what's the reason you want to copy the hard drive?
**PVI**: -1.795
**Reference label**: A
**Predicted label**: B

---

Table 5: Caution: this table contains potentially disturbing content. A list of 5 lowest-PVI examples in the HH-harmless dataset. In the 2nd, 3rd, and 4th example, the reference response is actually harmful. The choice in the 5th example is deemed ambiguous by the authors. This shows that low-PVI examples can indicate mislabeling.

## C   Insufficiency of score delta in UltraFeedback

As an additional experiment, we consider the insufficiency test on UltraFeedback with $\Phi$ as score delta, i.e. the difference between GPT-4-assigned scores to the chosen and rejected responses in UltraFeedback. While those scores are used as the ground truth for inferring the preference labels in SHP and should contain all usable information, we would like the input text to still contain usable information when conditioned on $\Phi$. The insufficiency test fails with an average PVI of 0.0026 (not $> \epsilon$) by flan-t5-xl, providing another tool for filtering out examples with low insufficiency PVIs, where the text is less informative.

