# OpenReview forum: "Data Checklist: On Unit-Testing Datasets with Usable Information"
_colmweb.org/COLM/2024/Conference — COLM_

### Official Review · Reviewer_z3Dx · 2024-04-22

**Rating:** 7
**Confidence:** 3
**Ethics Flag:** 1

**Summary:**

This paper presents a novel framework for evaluating the quality of datasets used to train languages models using an information-theoretic model. The authors describe this framework and how it can measure applicability, non-exclusivity, necessity and insufficiency of certain features and show application of this to NLI, hate speech and human preferences. They apply this framework to data filtering and show some improvements in task performance on smaller filtered datasets.

**Questions To Authors:**

How was $\espilon$ selected and what effect would changing it to 0.1 or 0.001 have?
Can you explain the V-Info scores in Table 2

**Reasons To Accept:**

Overall, this is an interesting paper looking at a specific challenge in language modelling that has not received sufficient attention. The paper is well-written and presents a flexible framework that could be applied to a lot of tasks in language modelling. The results seem to show effective improvements in data filtering showing the value of this work.

**Reasons To Reject:**

There are not really any baselines in this work, although I also am not sure what to compare this work too as well, as there are (surprisingly) few papers looking at the quality of data going into LMs. This recent [NeurIPS paper](https://proceedings.neurips.cc/paper_files/paper/2023/file/6b9aa8f418bde2840d5f4ab7a02f663b-Paper-Conference.pdf) may be quite relevant.
The first set of results in Section 4 are really interesting, but the conclusions are based on measuring relative to a value $\espilon$ that has been chosen arbitrarily. It seems strong to make claims such as in Table 2. I am also not sure of what the connection is between the "V-info" in Table 2 and the pass criterion.
The second set of results (Sec 5) are also very exciting, but are only on two datasets and don't seem to have significance analysis or comparisons to baselines.

## Minor
The references section has not been composed with the same care as the rest of the paper. There are captialisation errors, incomplete citations and references to arXiv for papers that have been published at peer-reviewed venues.

---

> ### Author Rebuttal · Authors · 2024-05-29
>
> Thank you for your review and valuable feedback.
>
> > Baseline papers that look at the quality of data going into LMs
>
> The discovery of annotation artifacts is typically evaluated qualitatively rather than quantitatively, even in well known papers like Dataset Cartography, since the existence of a particular attribute is a binary outcome that is hard to quantify in a granular way.
> We compare our findings with prior work on discovering data artifacts and show that our checklist can rediscover known artifacts.
>
> > Sec 5 has only two datasets and does not have enough analysis.
>
> Data filtering is indeed a very exciting area, but since the focus of this paper is to formalize the data checklist taxonomy,  data filtering is presented as a possible application—but not the main focus.
> However, we will consider experiments on other datasets, such as TinyStories for language modeling.
>
> Thank you for pointing us to the NeurIPS paper on data selection! We will make sure to include it as a reference.
>
> > How is the $\epsilon$ chosen and more clarification on table 2.
>
> As noted in Table 1, in each of the set of tests presented, we can choose either the $>\epsilon$ or $<\epsilon$ version as our assumption to test (chosen as viability, applicability, sufficiency, exclusivity, and necessity in Table 2).
> $\epsilon$ is the tolerance you have for test failure. Because we are ultimately taking an estimate of the V-info, even if the true V-info is 0, our estimate may be slightly above or below that. This is why $\epsilon$ is set to a small but non-zero value of 0.01. In practice, the user may also want to use values that are much larger than zero to satisfy some legal or product requirement (e.g. the gender attribute may be used to predict Y, but only to a certain extent, say 0.25 bits of information).
>
> The implications of the table 2 scores are explained on page 6, but to go into more detail, take the sufficiency test as an example; it asserts that “X contains no more than 0.01 bits of usable information about Y that is not already contained in the length of X”. The test score is computed as $I(X -> Y | length(X)) = H(Y| length(X)) - H(Y| length(X) U X)$.
> Since this score is 0.036, which is greater than 0.01, the test fails, giving us the implication that the length alone is not sufficient to predict Y.
>
> Nit: We will make sure to fix the references in the camera-ready.
>
> We hope this response offers some clarification and, if it does, we hope you consider increasing your score!

---

> > ### Comment · Reviewer_z3Dx · 2024-06-05
> > **Acknowledgement**
> >
> > Thanks for the explanations. Great work.

---

### Official Review · Reviewer_FFSv · 2024-05-07

**Rating:** 7
**Confidence:** 4
**Ethics Flag:** 1

**Summary:**

The paper adopts a V-information-based checklist to evaluate datasets. The authors argue that structured dataset audits should precede model behavior analysis. The proposed taxonomy starts with dataset viability, and then the other dataset feature-based measures take place, e.g., applicability, the selected feature is helpful for the downstream task. The authors frame the checklist as a tool to judge model efficacy and data efficiency. In particular, they show that one can filter some dataset instances using the appropriate V-information-based measure from the checklist without performance degradation on models. The experiments verify some undesired (the ratio of overlap between premise and hypothesis in the case of entailment classification) and desired (insufficiency of the profane words) artifacts in SNLI and hate speech detection datasets, respectively. They further support their claims with some V-information-related attributes (word complexity and response length) in text-to-text datasets, such as preference alignment.

**Questions To Authors:**

The title does not reflect the content precisely. "An Axiomatic Basis for Dataset Evaluation" would be an alternative.

**Reasons To Accept:**

A formalized approach for dataset verification.
Experiments verify the known artifacts in SNLI and hate speech detection datasets.
They also provide new artifacts (response length, word complexity, multilingual, etc.) aligned with V-information-based measures in preference alignment.

**Reasons To Reject:**

The authors claim that they provide a library for easy creation of checklists for text generation as well as classification. However, they do not give the library link for the practical evaluation.

---

> ### Author Rebuttal · Authors · 2024-05-29
>
> Thank you for your review and valuable feedback!
>
> > The authors claim that they provide a library for easy creation of checklists for text generation as well as classification. However, they do not give the library link for the practical evaluation.
>
> We did not include the link to avoid violating anonymity rules; we will release the code and include the link to the library in the paper after the anonymity period. Meanwhile, feel free to take a look at the supplementary material that we uploaded with the paper, which includes the code.
>
> > Q: The title does not reflect the content precisely.
>
> Thank you for your suggestion! We’ll consider the title you suggested.
>
> We hope this response offers some clarification and, if it does, we hope you consider increasing your score!

---

> > ### Comment · Reviewer_FFSv · 2024-06-04
> > **Acknowledgment**
> >
> > Dear Authors,
> >
> > Thank you for your response. I couldn't reach the supplementary material due to a technical issue. Providing an anonymized repository would be a good alternative. I already have a high score.
> >
> > Best,
> > FFSv

---

### Official Review · Reviewer_Vgca · 2024-05-11

**Rating:** 6
**Confidence:** 4
**Ethics Flag:** 1

**Summary:**

This paper proposes a data checklist. This is a taxonomy of unity tests based on the V-information. This is an interesting topic and worth studying.
Some parts are hard to follow due to the use of several abbreviations.
SNLI, RLHF, and NLU are not defined in this paper.
It is not clear to me how the 10 tests were defined. The paper just presents their formula without further description of the rationale behind each one.
The section Related Work should be relocated at the beginning of the text.

**Questions To Authors:**

For me, the paper misses a better understanding of each test domain. I mean, which range of values is expected for each test?

**Reasons To Accept:**

-	The paper mathematically formalizes all new propositions well.
-	The paper indeed presents three different contributions.

**Reasons To Reject:**

-	I think the paper needs more focus. It presents too many different “contributions” in the text. Perhaps it could be extended and published as a journal paper.

---

> ### Author Rebuttal · Authors · 2024-05-29
>
> Thank you for your review and valuable feedback!
>
> > The paper presents too many different “contributions”.
>
> We believe that the contributions we present in the paper are closely related. Building upon prior work in V-usable information, we propose a test taxonomy for checklisting datasets, and we then present experiments as examples of how people can utilize it, and show that this checklist is indeed useful. Finally, we extend this idea and take a step further to show how the checklist can be used for data filtering.
> If any one of these contributions were left out, it would become a purely theoretical or purely empirical paper, but it is a work that spans both.
> Given an additional page for the camera-ready, we will include more contextual and clarifying information to make the paper read more smoothly.
>
> > which range of values is expected for each test?
>
> Great question! We will make sure to clarify this in the paper for the camera-ready.
>
> Where |v| is the size of the vocabulary, the domain for all tests is [0, log |v|).
>
> V-information is formulated as the difference between two entropy values (negative log probabilities), $H_V(Y)$ and $H_V(Y|X)$, where $H_V(Y) >= H_V(Y|X)$  since having access to X should never make Y less predictable (due to the optional ignorance property of the model family V).
> Thus we have a lower bound of 0.
> The upper bound of $H_V(Y)$ is the entropy of a perfectly uniform distribution over tokens, which is $E_{y} [ - log f\[O\](y) ] = log |v|$.
>
> We hope this response offers some clarification and, if it does, we hope you consider increasing your score!

---

> > ### Comment · Reviewer_Vgca · 2024-06-05
> >
> > Thank you for the comments. I hope the paper canbe improved with all these new information.

---

### Official Review · Reviewer_W11a · 2024-05-15

**Rating:** 6
**Confidence:** 2
**Ethics Flag:** 1

**Summary:**

This paper proposes the concept of "data checklists", which are taxonomies of unit tests that formalize measures of usable information in a dataset. Although there are similar checklists for testing models, this appears to be the first attempt to develop checklists for the data itself, demonstrating the originality of the idea. The experiments carried out with various popular datasets for tasks such as toxic language detection and natural language inference show the utility of the approach. While the discussion of the methodology seems clear, the paper overall is not particularly accessible to readers lacking in-depth knowledge of popular datasets, which detracts from its value.

**Questions To Authors:**

On page 6, you mention that word complexity might impact preference, but you never define word complexity. I recommend explaining exactly what is meant, perhaps with examples of words with low and low complexity.

**Reasons To Accept:**

* The idea is original. While there are model checklists, there does not appear to be anything similar for data. This paper proposes both a new approach and a new target.
* The authors' use of V-information for sequences is a novel contribution.
* The proposed framework, formalization, and software can offer utility to researchers who plan to use an existing dataset for a new task.

**Reasons To Reject:**

* The paper assumes a lot of knowledge on the part of the reader about datasets and training techniques. Every dataset is initially referred to as an acronym with no explanation of the content or purpose of the dataset until much later in the paper, if ever (e.g., SNLI, HH-harmless, SHP). (The one exception is DWMW17.) Similarly, many training techniques are never expanded or explained (e.g, RLHF, DPO). I know there are lots of people who are already familiar with all the datasets and all the training techniques, but a person from outside the LLM space who are interested in using the library to learn more about the relevance particular dataset for a new task might really struggle to get out the information they need from the paper.
* The discussion of prior work (which I think should be moved to the front of the paper) talks a lot about model checklists, with only brief references to prior work on finding dataset artifacts, the focus of the paper. I feel like the prior work that is directly related could be highlighted more to provide more context for the work.
* While the results show that the approach is doing something, there is no comparison to any other approach. I know this is a new idea, but is there any prior work that does something similar that could be considered?

---

> ### Author Rebuttal · Authors · 2024-05-28
>
> Thank you for your review and valuable feedback!
> > The paper assumes a lot of knowledge.
>
> Due to the page limitation, we weren’t able to include detailed background information on all the datasets and preference alignment techniques mentioned in the paper, and instead, we provide brief context and citations for unfamiliar readers.
> However, we recognize that a smoother introduction and more explanations of datasets and techniques will be particularly helpful among a larger audience beyond the LLM space, and will facilitate the understanding and adaptation of our method. We will make sure to accommodate this suggestion in the camera-ready version.
>
> > The discussion of prior work talks a lot about model checklists instead of finding dataset artifacts, the focus of the paper.
>
> The major part (first 2 of the 3 paragraphs) of Related Work (Section 6) is indeed focused on dataset analysis and annotation artifacts, where we cite many prior works on data attribution, data curation, spurious patterns, etc.
>
> In addition, we are promoting the direct evaluation of datasets themselves instead of tracing problems back from model behavioral evaluation, which is why we start the intro by talking about model checklists and comparing them with our method.
>
> > While the results show that the approach is doing something, there is no comparison to any other approach.
>
> The discovery of annotation artifacts is typically evaluated qualitatively rather than quantitatively, even in well known papers like Dataset Cartography.
> Although there is no agreed-upon list of annotation artifacts in the literature, we compare our findings with prior work and show that we can rediscover known artifacts with the new framework (section 4.1). The response length artifact in preference datasets has also been observed in prior work (Singhal et al., 2023), and we formalize the observation with our checklist.
> Our new taxonomy builds upon prior work, and we present contributions from related methods in the paper, even though a direct comparison is not easily available.
>
> > Q: You never define word complexity.
>
> We measure word complexity as the frequency of occurrence in a large corpus (as mentioned in that section), where less frequent words are regarded as being more complex. For greater clarity, we will use the term rarity instead.
>
> Thanks again for your questions and comments. We hope this response offers some clarification and, if it does, we hope you consider increasing your score!

---

> > ### Comment · Reviewer_W11a · 2024-06-04
> > **thank you**
> >
> > Thank you for your response, authors!
> >
> > * The page limitation is not really an excuse for not providing enough information for people to understand the paper. Part of writing papers is figuring out how to work within the page limit, which I hope you will do. :)
> >
> > * I would move the related work to the beginning of the paper. It is difficult to understand the strength and the context of the contribution.
> >
> > * Ah yes, complexity is definitely not what I thought it was! I think it would be helpful to rename it and explicitly define it in the paper.

---

### Decision · Program_Chairs · 2024-07-10

**Decision:**

Accept

**Comment:**

This paper proposes a “data checklist” for unit-testing datasets to identify artifacts and ensure the data aligns with model behavior, using a taxonomy based on V-information literature.

Strengths:
1. Novelty: Reviewers all noted that there has been surprisingly few papers looking specifically at dataset quality, and they appreciated the concept of data checklists for datasets (Reviewer W11a, Vgca).
2. The research artifacts (frameworks and tools) could be beneficial (Reviewer W11a).
3. The research is well conducted, including mathematically well-formulated propositions (Reviewer Vgca) and sufficient verification of known dataset artifacts (Reviewer FFSv).

Weaknesses:
1. The paper used too many acronyms without providing enough background contexts on the datasets and technical methods, which make it a bit difficult to read (Reviewer W11a)
2. There is a lack of comparison with prior work, though reviewers did acknowledge that there tend to be less related work in the field (Reviewer W11a, Vgca, z3Dx)
3. Needs better explanation and definition of terms like word complexity and more context for the tests (Reviewer W11a, Reviewer Vgca).

In their rebuttal, the authors added clarifications to the notations used in the paper, and clarified how they compare to prior work (e.g., discovering dataset artifacts). Their final ratings are all above the acceptance ratio. That said, some reviewers still strongly encouraged the authors to release the code and revise the writing.

I believe the paper presents a significant contribution to the field with its innovative approach to dataset evaluation, and the presentation issues can be fixed after acceptance.